# Evaluating and Enhancing the Preparation of Patients and Families before Pediatric Surgery

**DOI:** 10.3390/children7080090

**Published:** 2020-08-05

**Authors:** Christopher Newell, Heather Leduc-Pessah, Lisa Bell-Graham, Nivez Rasic, Kerryn Carter

**Affiliations:** 1Cumming School of Medicine, University of Calgary, Calgary, AB T2N 4N1, Canada; hlleducp@ucalgary.ca; 2Alberta Children’s Hospital, Calgary, AB T3B 6A8, Canada; lisa.bell-graham@albertahealthservices.ca; 3Alberta Children’s Hospital Research Institute, Calgary, AB T3B 6A8, Canada; nivez.rasic@albertahealthservices.ca; 4Section of Pediatric Anesthesiology, Department of Anesthesiology, Perioperative and Pain Medicine, University of Calgary, Calgary, AB T3B 6A8, Canada; kerryn.carter@albertahealthservices.ca

**Keywords:** pediatric surgery, patient education, patient-centered care, pre-operative anxiety

## Abstract

Surgery can be a difficult and unfamiliar experience for children and their families. We examined the ability of existing information to help families feel better prepared for surgery at the Alberta Children’s Hospital (ACH) and evaluated the best way to enhance its content and accessibility. We developed an online survey for families who have had surgery at ACH. Participants were recruited through pre-existing patient networks and from the ACH Short Stay Unit (SSU) between October 2018 and October 2019. The survey asked participants to evaluate the information available to prepare them for surgery and requested suggestions for improvement. Our survey results show that those who completed the in-person Surgery 101 program felt significantly more prepared for surgery. Of those who did not attend; 40% would have been interested in participating but were unaware that the program existed; and 17% planned to attend but were unable to; due to work or travel distance. Participants felt additional resources via online content or paper handouts would be most valuable. We used this information to prepare an online accessible summary of the Surgery 101 program and tour in the form of a video to reach more Albertan families preparing for surgery for their children

## 1. Introduction

More than one million patients undergo a surgical procedure in Canada annually [1], with at least 64,000 being pediatric cases [2]. Surgery can be a difficult and unfamiliar experience for both children and their families, which can be influenced by parental and child stress, anxiety, and catastrophizing [3]. Data suggests that up to 60% of pediatric patients experience high levels of pre-operative anxiety [4], which can lead to increased levels of post-operative pain, analgesic consumption, and general anxiety [5].

To address these concerns, many hospitals provide access to pre-operative resources aimed at alleviating pre-operative stressors. Existing information to help prepare families for surgery at the Alberta Children’s Hospital (ACH) includes paper handouts, the ACH website, the Family and Community Resource Centre (FCRC), and the Surgery 101 program. The FCRC is a dedicated service within ACH which aims to provide credible health information, family support, and resources to the surrounding community. Surgery 101 is a specialized program run by Child Life Specialists at ACH providing in-hospital sessions to orient patients and families on what to expect when preparing to undergo surgical intervention.

Internal ACH data report that approximately 10,000 surgeries take place annually. The existing Surgery 101 program receives approximately 50 families per year, reaching only 0.5% of potential patients. We hypothesized that the existing Surgery 101 program is valuable in preparing families for surgery but that its accessibility and enrollment may be improved upon. Therefore, we developed an online survey for families who have had surgery at ACH. Participants were asked to evaluate the information that was available to prepare them for surgery and specifically highlighted participant awareness of the Surgery 101 program. The study presented here was designed to improve the surgical experience for pediatric patients and their families from across Alberta who were coming to ACH for surgery and may serve as an example to other hospitals caring for children.

## 2. Materials and Methods

### 2.1. Study Design, Setting, Survey Participants and Size

This study employed a retrospective cohort study design using an online survey. Designed to be completed by patients and/or their parents, this open survey was comprised of 24 questions or 30 questions if they had participated in Surgery 101. For a full list of survey questions, see Appendix A. A 10-point Likert scale was used for ranking. All survey questions were reviewed and approved by all members of the research team. No incentives were offered for completion of the survey. The survey was conducted through the Alberta Health Services approved online survey platform (Select Survey). Patients and/or their parents were asked to provide consent online through Select Survey and all consent documents were provided through the Select Survey platform. The site was not password protected but a link or information with the survey ID was helpful to locate it. Since families may have multiple children or experiences, we did not limit responses by IP address or computer, and we asked families to report only their most recent experience. At initiation of the survey, participants were provided with a Select Survey code through which they could modify or delete their responses at any time.

We surveyed pediatric patients, age 0–18 years, who have had surgery at ACH and parents of pediatric patients who have had surgery at ACH. Past participants of the Surgery 101 in-hospital program during the past two years (2017–2019) were contacted via e-mail. Participants who did not participate in Surgery 101 were recruited through the ACH Patient and Family Centered Care Network (e-mail list of families willing to receive information about research), the Day Surgery Unit at ACH (poster and optional access to survey link), and through approved posters displayed throughout ACH. Although general demographic data were collected, patient or family names and contact information were not collected and all data was anonymized upon receipt by the investigators. Only anonymized data were stored on the secure Select Survey platform.

A sample size of 96 participants was calculated prior to survey implementation using a population of 10,000 pediatric surgical cases per year at ACH, a margin of error of 10%, a 95% confidence interval, and a response distribution of 50:50. This study was carried out in accordance with the regulations of the University of Calgary’s Conjoint Health Research Ethics Board and the Declaration of Helsinki (REB18-0272).

### 2.2. Thematic Analysis

A qualitative deductive thematic analysis was conducted of the free text reported in the survey. The global themes from this data were separated into Surgery 101 specific feedback and general feedback. The most common re-occurring codes were identified on qualitative analysis of the text and grouped into themes. Codes and themes are reported in Figure 1. Direct participant comments are anonymously reported within the text.

### 2.3. Statistical Analysis

Statistical analysis was performed using GraphPad Prism for Windows, Version 6.0.7 (GraphPad Software Inc., La Jolla, CA, USA). Differences between groups were determined by Student’s *t*-tests, where *p* < 0.05 was significant. Data are expressed as mean ± SEM.

## 3. Results

Participant recruitment occurred between October 2018 and October 2019, with each participant being represented only once (Table 1). A total of 140 responses to the survey were collected, of which 97 were completed and used for analysis. Four of the responses were completed by the patient and the remaining 93 by a parent. Hereafter, patient and/or parent survey responses will be referred to as participants. A total of 32 (33.0%) survey participants completed the Surgery 101 program and 65 (67.0%) participants did not complete Surgery 101. When examining all responses, the age of participants ranged from <1 to 16 years of age. Males comprised 60.8% of participants. Of surgical patients, 64.9% spent at least one night in the hospital. The three most common types of surgical procedures were ENT, GI, and Urology/Gynecology. The amount of time between surgery being planned and the surgical intervention was typically >28 days (55.7%); however, Non-Surgery 101 participants more frequently experienced <7 days’ notice (20.0% vs. 6.2%, respectively). Most participants travelled from within Calgary to attend their surgical procedure (74.3%). A complete comparison of patient demographic and clinical characteristics between Surgery 101 and Non-Surgery 101 groups is described in Table 2.

### 3.1. Surgery 101 Better Prepares Families for Surgery

Surgery 101 participants self-reported that, both as a patient and as a parent, they felt more prepared for surgery (Figure 2A). When asked to subjectively rate their overall level of preparedness for surgery, Surgery 101 participants reported they felt significantly more prepared than Non-Surgery 101 participants (Figure 2B, *p* = 0.0113). Collectively, these data demonstrate that the Surgery 101 program is effective at making families feel better prepared for surgery. All Surgery 101 participants had positive comments about their experience and commented both broadly and specifically on the positive impact of the Surgery 101 session and the expertise of the Child Life Specialists who ran the sessions. Many parents reported that the program helped themselves and their children feel more prepared for surgery, consistent with the data reported in Figure 2A. In addition, many parents reported that they felt the session was appropriately geared towards children. Specific comments that support this theme include: “It is a great program and it helped both my daughter and I prepare.”, “…the leader of Surgery 101 did an amazing job. She spoke to the kids on a level they could understand and answered all their questions.”, and “Overall it was a very good experience. We felt prepared…”. Overall, the thematic analysis of feedback from Surgery 101 participants supported the conclusion that the in-hospital Surgery 101 session allows families to feel better prepared for surgery.

### 3.2. Surgery 101 Is Not Widely Accessible to Potential Participants

Of the 97 survey participants, 33.0% attended Surgery 101 (Figure 3A). The remaining participants can be sub-divided into those who reported interest in attending the session but were unable to attend for various reasons (38.1%) and those who were not interested in attending Surgery 101 (28.9%) (Figure 3A). Those who did not attend Surgery 101 were further divided based on their reason for not attending (Figure 3B). Of those who reported interest in attending, the majority (40.0%) expressed that they were unaware of the Surgery 101 program (Figure 3B). The remainder reported that they were unable to attend due to their distance from the hospital or to scheduling conflicts (Figure 3B). Of those who attended Surgery 101, the majority (77.8%) had to take time off work to do so, which is an identified barrier to physically attending Surgery 101. Most participants who reported they were not interested in attending Surgery 101 were also unaware of the program and only 7.7% of those who did not attend Surgery 101 knew about the program and made the choice to not attend (Figure 3B). Overall, 57% of families who did not attend the Surgery 101 program reported that they were interested in attending but were unable to for various reasons. These data provide evidence that many families who may have benefitted from the Surgery 101 program were unable to utilize this resource.

### 3.3. Accessibility of the Information in Surgery 101 Can be Improved

We asked survey participants to select which methods of communication they would find most beneficial to help them prepare for surgery. Participants were able to select multiple options; thus, the number of responses exceeds 97 (Figure 4A). The most desired methods of communication were paper handouts and a website, followed by a smartphone application (app) (Figure 4A). Since ACH has an existing website with the option of including additional information and paper handouts are already in circulation, we specifically surveyed the likelihood of families to download a smartphone app designed to prepare their child for surgery. Responses varied widely from 1 (not at all likely) to 10 (very likely), with a mean of 6.5 on a Likert scale (Figure 4B). We initially predicted that a smartphone app would be much more desirable to this population; however, given these results, we opted to pursue other means of improving the accessibility of the Surgery 101 program. Considering paper handouts and the website are already in place, we sought to further analyze the qualitative data to better understand what the reported benefits of the Surgery 101 program are and how these could bolster existing resources.

For participants who attended the Surgery 101 session, the most resounding theme was hospital orientation and what to expect on the day of surgery. Parents reported that the tour of ACH and visual reinforcement regarding what would be happening and where they and their child would be on surgery day is what made them feel the most prepared. Some illustrative examples include: “Walking the halls before the surgery, seeing the unit(s) was all extremely helpful to know what visually would be happening the day of surgery” and “Going through all of the steps we would go through the day of was very helpful. To have us and our daughter know where she would go throughout the day gave us comfort.” Therefore, based on these specific comments, the themes from our survey participant responses, and the inability to accommodate site tours for all interested participants, our research team concluded that a video walkthrough would be the best way to extend the impact of the Surgery 101 program. We created a video depicting a family’s experience on surgery day, which is now available on the ACH website (https://youtu.be/RQ5Xl6EDOlw). We believe that this video will be one of the first steps in improving patient and family’s experiences and reducing pre-operative anxiety.

Although not the primary objective, our thematic analysis also generated data on areas for improvement both within the Surgery 101 program and for the overall surgical process at ACH (Figure 1). Identified as the main areas of improvement were general post-surgical care, surgery-specific care, and resources for families within the day surgery unit. The possibility of including post-surgical care in a smartphone app remains a future goal.

## 4. Discussion

We found that the in-hospital Surgery 101 program improves family and child preparedness for surgery at ACH and is well regarded by those who have attended it. With the evolution of patient-centered care, there has been a major shift to promote education and patient empowerment in the decision-making process. In pediatric medicine, parental education has traditionally been a focus. However, there is tremendous value in incorporating educational elements, specifically for the child. We identified surgery as an area where family education would be beneficial and effective due to the high degree of stress and anxiety [4]. In addition, evidence has shown that patient education prior to surgery is limited [6,7].

Given that most surgeries are scheduled in advance [8], there is time to improve patient and family education pre-operatively, improving the perioperative experience. Several existing studies have explored specific components of pediatric surgical preparation, including assessing different modalities for providing information [9], anesthesiology-specific preparation [10] as well as the impact of pre-operative anxiety [11]. We specifically sought to assess the impact of an in-person orientation session and collect feedback from families at ACH. Our quantitative and qualitative results support the benefit of a program such as Surgery 101 to both parents and patients. This information guided our decision to develop additional resources to reach more families; however, it also suggests that similar programs would be beneficial at other Canadian Pediatric Hospitals.

The major barriers identified to accessing Surgery 101 were a lack of awareness, difficulty with access due to parental work schedules, and physical distance from hospital. To address the lack of awareness, we examined the existing Surgery 101 paper handouts with the Child Life Specialists at ACH and determined that these pre-operative packages were reaching all clinics in the hospital. We also learned that a revised website was underway for ACH. Therefore, we chose to develop a novel resource that could be electronically distributed and could be used to enhance the new ACH website. This was an online video that provides a walkthrough of what patients and families are to expect when travelling to ACH for surgery (https://youtu.be/RQ5Xl6EDOlw). This video version covers key aspects of the Surgery 101 program and confers many advantages. Firstly, it provides a cost-effective alternative to families who were not able to attend Surgery 101 due to distance from the hospital or who needed to take time off from work. Secondly, an electronic resource can be more easily disseminated and may therefore reach families who were not aware of the Surgery 101 program. Finally, this video provides advantages for specific sub-populations of patients. A hypothesis when initiating this study was that additional resources would positively impact children with high levels of anxiety or with existing conditions such as Autism Spectrum Disorder. This theme was also identified by survey participants; “[My son] is not an overly anxious child but for children that are, [Surgery 101] would be a must.” We do not anticipate that an online video will discourage families from attending the in-hospital Surgery 101 program but believe it will provide an added benefit to review the Surgery 101 program at their leisure. Child Life Specialists also reported that older patients typically do not feel comfortable attending Surgery 101 and our data show that the oldest patient to attend Surgery 101 and complete our survey was 12 years old. Child Life offers one-on-one Surgery 101 sessions for teenagers; however, an online video is an additional resource that teenagers can access without undue social stressors.

We predicted that a smartphone app would be more desirable to families preparing for surgery. Some of the concerns with the use of a smartphone app included the need to find and download the app for one-time and short-term use, the need to develop it across multiple platforms, and the utility of an app targeted to one centre. Since our thematic analysis identified a major theme of being able to visually see where to go on the day of surgery, we felt it was appropriate to focus on an ACH-specific resource. We found that a variety of generic (non-site specific) applications or online videos are available for surgical preparation [12,13,14] and we are continuing to explore options for further development of a site-specific app. A strength of our study was the ability to directly compare a family’s sense of preparedness for surgery with and without accessing the in-person Surgery 101 program. This allowed us to assess the Surgery 101 content as additional valuable content to make more widely accessible.

Some limitations of our study included a limited sample size as well as targeted recruitment through our e-mail list to families who had agreed in advance to participate in research and may be more involved with the hospital. However, our posters and recruitment in the day surgery area reached all families. Since the study was voluntary, those who chose to complete it may have been more likely to require additional preparation for surgery or have additional perioperative anxiety and may have been more likely to spend longer times in the hospital contributing to our high number of patients staying >24 h. The survey tool used in this study was not previously validated and may have benefited from specific questions regarding the development of a video. In addition, since 19.6% of participants completed the survey >12 months after their surgery date, their responses may have been impacted by recall bias. We were also not able to collect an equal distribution of Surgery 101 and non-Surgery 101 participants, as indicated in our statistical power calculation.

To best improve the accessibility of Surgery 101 and reach families who are unable to attend the in-hospital session, we have created a video summary of surgery day, which is available on the ACH website. We believe that this video, as well as ongoing efforts to improve family preparation prior to surgery, will enhance the experiences of patients and families undergoing surgery.

## Figures and Tables

**Figure 1 children-07-00090-f001:**
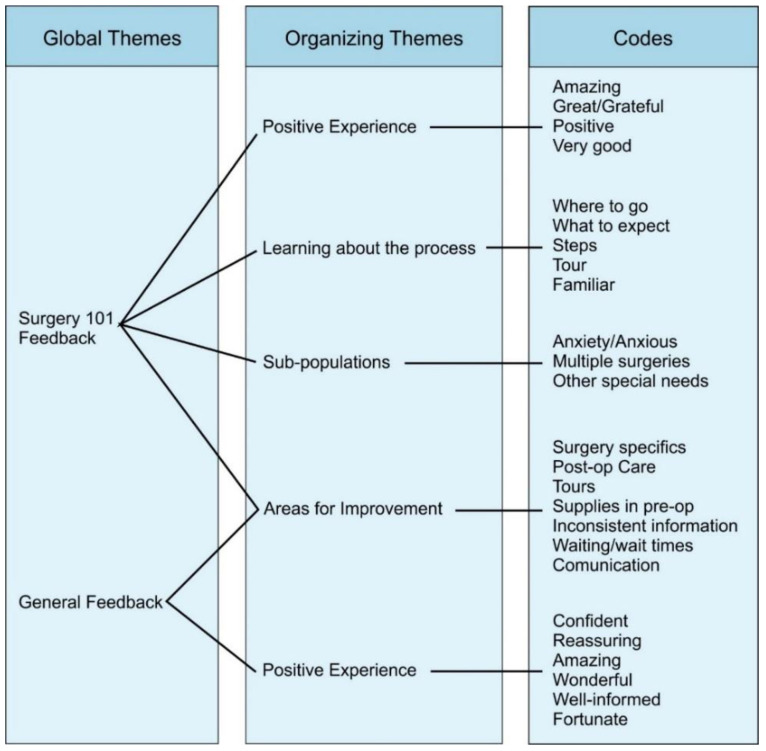
Thematic Analysis Summary of Participant Free Text Responses. A qualitative deductive thematic analysis was conducted of the free text participants responses to the questions (1) “Is there any further information you wish to share regarding your and/or your child’s experience at Alberta Children’s Hospital?”, (2) “What was something you learned from the Surgery 101 Program?” and (3) “What was something you wished you learned from the Surgery 101 Program?” All responses were analyzed into global themes and organizing themes with code words for each organizing theme displayed here.

**Figure 2 children-07-00090-f002:**
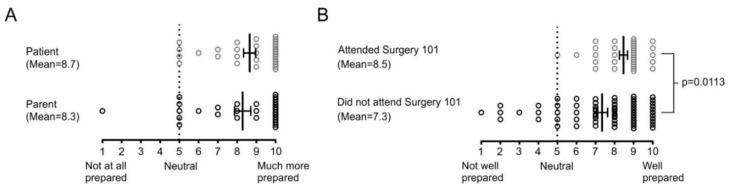
Surgery 101 Better Prepares Families for Surgery. (**A**) Responses of survey participants who attended Surgery 101 to the question “Did you feel more prepared for your child’s surgery after attending Surgery 101?” on a Likert scale where 1 represented “Not at all Prepared” and 10 “Much more prepared”. Participants responded on behalf of themselves and their child, or their parent if completed by the patient. Individual data points as well as mean ± SEM are shown. (**B**) Responses of all survey participants to the question “Overall, did you feel well prepared for your child’s surgery?” on a Likert scale where 1 represented “Not Well Prepared” and 10 “Well Prepared”. A two-tailed unpaired t-test comparing Surgery 101 participants vs. Non-Surgery 101 participants was conducted (*t* = 2.583, *df* = 94, *p* = 0.0113). Individual data points as well as mean ± SEM are shown.

**Figure 3 children-07-00090-f003:**
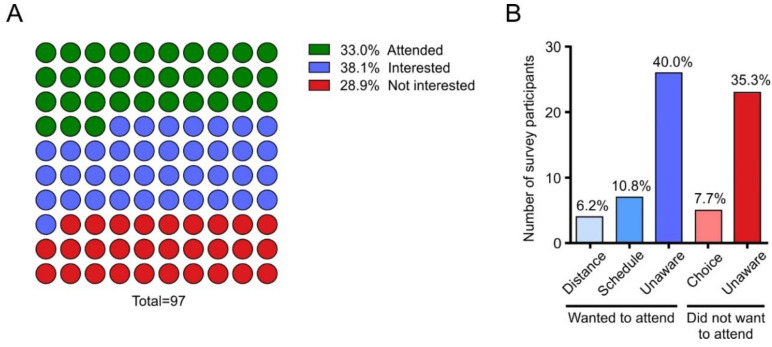
Participation in the Surgery 101 Program. Responses of all survey participants to the question “Did you participate in the Surgery 101 program?” where responses included options that specified why they were unable to attend, whether they were aware of the program, and whether they would have wanted to attend or not had they been aware. (**A**) Percent of a whole plot displaying the compiled responses of all survey participants into “Attended”, “Did not attend but interested in attending” and “Did not attend but not interested in attending”. The responses are represented as a percentage of all survey participants (*n* = 97). (**B**) Responses only from participants who did not attend the Surgery 101 program further broken down into reasons for not attending. The responses are represented as a percentage of those who did not attend (*n* = 65).

**Figure 4 children-07-00090-f004:**
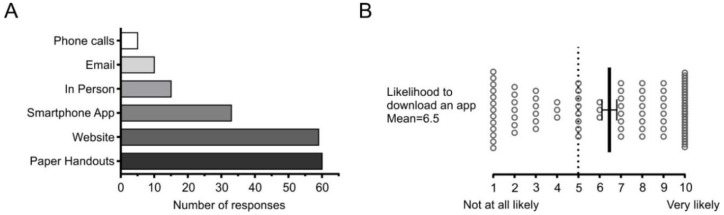
How Participants Prefer to Receive Information Regarding Preparation for Surgery. (**A**) Responses of survey participants to the question “Through what form(s) of communication would you find it easiest to prepare for your child’s surgery?” where participants were able to select as many options as they wanted. Thus, the total number of responses to this question (*n* = 182) exceeds the number of survey participants. (**B**) All survey participants’ responses to “Would you be likely to download an app that provided information and resources to help prepare for surgery?” on a Likert scale where 1 represented “Not at all Likely” and 10 “Very Likely”. Individual data points as well as mean ± SEM are shown.

**Table 1 children-07-00090-t001:** Patient demographics and clinical characteristics.

Characteristic	*n* (%)
*Participants*	97
*Sex (%)*	
Male	59 (60.8)
Female	38 (39.2)
*Type of surgical procedure (%)*	
Dental	4 (4.1)
ENT	25 (25.8)
GI	16 (16.5)
Neurosurgery	2 (2.0)
Ophthalmology	7 (7.2)
Orthopedics	9 (9.3)
Plastics	5 (5.2)
Thoracic	1 (1.0)
Urology/Gynecology	28 (28.9)
*Age at operation (years)*	
Mean	5.8
Median	5
Range	<1–16
*Length of hospital stay (%)*	
<24 h	34 (35.1)
>24 h	63 (64.9)
*Time notified before surgery (%)*	
<7 days	15 (15.4)
7–28 days	28 (28.9)
>28 days	54 (55.7)
*Distanced travelled for procedure (%)*	
Within Calgary	72 (74.3)
<100 km	17 (17.5)
100–250 km	6 (6.2)
>250 km	2 (2.0)
*Time elapsed since surgery (%)*	
<1 month	35 (36.1)
1–6 months	32 (33.0)
6–12 months	11 (11.3)
>12 months	19 (19.6)

**Table 2 children-07-00090-t002:** Separated Surgery 101 and Non-surgery 101 patient demographics with clinical characteristics.

Characteristic	Surgery 101 *n* (%)	Non-Surgery 101 *n* (%)
*Participants*	32	65
*Sex (%)*		
Male	23 (71.9)	36 (55.4)
Female	9 (28.1)	29 (44.6)
*Type of surgical procedure (%)*		
Dental	1 (3.2)	3 (4.6)
ENT	10 (31.2)	15 (23.1)
GI	4 (12.5)	12 (18.5)
Neurosurgery	0 (0)	2 (3.1)
Ophthalmology	4 (12.5)	3 (4.6)
Orthopedics	2 (6.2)	7 (10.8)
Plastics	1 (3.2)	4 (6.1)
Thoracic	0 (0)	1 (1.5)
Urology/Gynecology	10 (31.2)	18 (27.7)
*Age at operation (years)*		
Mean	5.5	5.9
Median	5	4
Range	1–12	<1–16
*Length of hospital stay (%)*		
<24 h	13 (40.6)	21 (32.3)
>24 h	19 (59.4)	44 (67.7)
*Time notified before surgery (%)*		
<7 days	2 (6.2)	13 (20.0)
7–28 days	8 (25.0)	20 (30.8)
>28 days	22 (68.8)	32 (49.2)
*Distanced travelled for procedure (%)*		
Within Calgary	25 (78.1)	47 (72.3)
<100 km	4 (12.5)	13 (20.0)
100–250 km	3 (9.4)	3 (4.6)
>250 km	0 (0)	2 (3.1)
*Time elapsed since surgery (%)*		
<1 month	11 (34.3)	24 (36.9)
1–6 months	14 (43.7)	18 (27.7)
6–12 months	6 (18.8)	5 (7.7)
>12 months	1 (3.2)	18 (27.7)

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
