# Peer review of "Evaluating and Enhancing the Preparation of Patients and Families before Pediatric Surgery"

_children, 2020, doi:10.3390/children7080090_

Round 1

Reviewer 1 Report

A well and fluently written paper of interest to pediatric surgeons. The paper adds no new information to what already documented in the literature. The subject har been discussed in couple of dissertations during the last twenty years providing similar results. The results are not unexpected. The scientific procedure is sound and well described. I miss no publications from the list of references.

I suggest that the authors start the discussion by stating their results and then discuss their results in the light of what is found in the literature. Furthermore, the authors should discuss the strength and weakness of their study.

Author Response

Thank you for your comments that the paper is “of interest to pediatric surgeons” and that “the scientific procedure is sound and well described”. We appreciate your suggestions and feedback and have revised the manuscript accordingly. Specifically, we have added a re-iteration of the results at the beginning of the discussion and have presented them in the context of the existing literature. In addition, we have added to the discussion the strengths and weaknesses of the paper.

Reviewer 2 Report

Surgical techniques are becoming more and more advanced nowadays. Therefore a sound explanation and preparation of patients and families to surgery is crucial. It also has legal implications, because a proper understanding of the planned procedure/surgery is necessary to obtain informed consent! In different countries, the approach to informed consent may vary, but the elements of informed consent remain the same - decision-making capacity, disclosure, comprehension and voluntariness. The authors have proposed a valuable tool that evaluates and enhances the process of preparation for surgery.

Author Response

Thank you very much for your comments regarding our paper. We appreciate your note that the paper proposes “a valuable tool that evaluates and enhances the process of preparation for surgery”.

Reviewer 3 Report

Overall the idea of examining preparation toward surgery is extremely important and timely and is an issue that most hospitals should be exploring. This topic is relevant to all children receiving surgery, especially those with histories of anxiety. However, the authors need to address the following areas:

1. Why did the authors make the anxiety questionnaire optional?  How was this info used? This topic was not discussed at all in the manuscript. Children and parents with anxiety may have responded differently to the preparation. If this is not a significant portion of the paper, I take out references to this questionnaire unless they are adequately addressed.  

2 It is not helpful to note that all ages of children could have potentially responded to the questionnaire.  Why was there not a cutoff for the youngest age of participant that could report on their experiences? Moreover, it appeared that only 4 children out of 93 reported on their experiences.  I would not include this data because it is not enough analyze comparisons between parents and children, etc. 

3. Why were only 97 out of 140 survey completed? Please clarify. 

4. The authors conclude in both the results and discussion that the Surgery 101 program s effective in preparing families for surgery. However, their data to support this is only perceptions of families and they did not look at whether behaviors were different (better coping with surgery, shorter hospital stay, etc.). The authors need to be careful about not making overgeneralizations based on the data. They can conclude that families FEEL better prepared, which is significant in itself.

5. The authors conclude that making a video to orient families around surgery is the best approach to addressing the concerns about the Surgery 101 program but it felt like an Ad Hoc decision not supported by data. I would feel more comfortable with this conclusion if they actually asked families about their interest in a video.

Overall, I feel like the topic the authors addressed is extremely important and the data collected is very helpful but there were overgeneralizations to the data and conclusions that are not supported by data.  A significant rewrite to the article is needed to address these issues.  

Author Response

Response to Reviewer 3

We thank Reviewer 3 for their positive feedback and constructive comments regarding our manuscript. Our responses to individual comments are listed numerically.

  1. The authors agree that the reference to the anxiety questionnaire does not bolster the strength of the current manuscript. We have therefore removed reference to this document. Our initial goal was to have the anxiety questionnaire act as a subsequent data point, although we were unable to collect enough representative data from our survey cohort.
  2. There was no designated age cut-off for completion of our survey as the Reviewer described. The authors organized our survey in this manner to allow the survey participants to dictate their own capacity to complete the survey with or without the need for parental/adult involvement. This decision was in keeping with the lack of a medical age of consent in pediatrics. Furthermore, the children who completed our study (4/97) were answering the same questions as the parents which is why this data was analyzed together.
  3. Only 97 responses were completed due to the optional nature of our survey questions which was in accordance with our Ethics proposal.
  4. The authors agree that the language surrounding these conclusions is something that needs to be carefully considered. We have altered the language throughout our manuscript to reflect that this data is qualitative in nature and generated from patient self-reported information (Example: lines 138-142).
  5. The authors have more clearly indicated the rationale for completing a video walk through of the hospital based solely on patient and parent reported testimony (lines 223-231).

Round 2

Reviewer 3 Report

Thank you for addressing the comments satisfactorily.

Author Response

Thank you for helping us to improve the impact of our manuscript.